# Molecular Characterization of *Staphylococcus aureus* Isolated from Chronic Infected Wounds in Rural Ghana

**DOI:** 10.3390/microorganisms8122052

**Published:** 2020-12-21

**Authors:** Manuel Wolters, Hagen Frickmann, Martin Christner, Anna Both, Holger Rohde, Kwabena Oppong, Charity Wiafe Akenten, Jürgen May, Denise Dekker

**Affiliations:** 1Institute of Medical Microbiology, Virology and Hygiene, Universitiy Medical Center Hamburg-Eppendorf (UKE), 20251 Hamburg, Germany; m.wolters@uke.de (M.W.); mchristner@uke.de (M.C.); a.both@uke.de (A.B.); Rohde@uke.de (H.R.); 2Department of Microbiology and Hospital Hygiene, Bundeswehr Hospital Hamburg, 20359 Hamburg, Germany; frickmann@bnitm.de; 3Institute for Medical Microbiology, Virology and Hygiene, University Medicine Rostock, 18057 Rostock, Germany; 4Kumasi Centre for Collaborative Research in Tropical Medicine (KCCR), Kumasi, Ghana; oppong@kccr.de (K.O.); danquah01@yahoo.co.uk (C.W.A.); 5Tropical Medicine II, Universitiy Medical Center Hamburg-Eppendorf (UKE), 20251 Hamburg, Germany; j.may@uke.de; 6Department of Infectious Disease Epidemiology, Bernhard Nocht Institute for Tropical Medicine (BNITM), 20359 Hamburg, Germany; may@bnitm.de; 7German Centre for Infection Research (DZIF), Hamburg-Lübeck-Borstel-Riems, 38124 Braunschweig, Germany

**Keywords:** rural Ghana, molecular epidemiology, chronic wounds, *Staphylococcus aureus*

## Abstract

Background: Globally, *Staphylococcus aureus* is an important bacterial pathogen causing a wide range of community and hospital acquired infections. In Ghana, resistance of *S. aureus* to locally available antibiotics is increasing but the molecular basis of resistance and the population structure of *S. aureus* in particular in chronic wounds are poorly described. However, this information is essential to understand the underlying mechanisms of resistance and spread of resistant clones. We therefore subjected 28 *S. aureus* isolates from chronic infected wounds in a rural area of Ghana to whole genome sequencing. Results: Overall, resistance of *S. aureus* to locally available antibiotics was high and 29% were Methicillin resistant *Staphylococcus aureus* (MRSA). The most abundant sequence type was ST88 (29%, 8/28) followed by ST152 (18%, 5/28). All ST88 carried the *mecA* gene, which was associated with this sequence type only. Chloramphenicol resistance gene *fexB* was exclusively associated with the methicillin-resistant ST88 strains. Panton-Valentine leukocidin (PVL) carriage was associated with ST121 and ST152. Other detected mechanisms of resistance included *dfrG*, conferring resistance to trimethoprim. Conclusions: This study provides valuable information for understanding the population structure and resistance mechanisms of *S. aureus* isolated from chronic wound infections in rural Ghana.

## 1. Introduction

*Staphylococcus aureus* is an important bacterial pathogen in all parts of the world, causing both community and hospital acquired infections. In particular methicillin-resistant *S. aureus* (MRSA) has evolved as a global health threat due to its resistance to beta lactam and other classes of antibiotics [1]. In the last 20 years the prevalence of MRSA appears to be increasing in many African countries as suggested by data from the first decade of the present century [2]. More recent reviews indicate ongoing epidemiological relevance of this resistance type in Africa [3], with increased reporting of outbreak-association in Western African Ghana [4]. In Ghana, the abundance of MRSA in carriage studies or clinical samples demonstrated large geographical differences [5,6,7]. Moreover, resistance of *S. aureus* to a variety of other locally available oral antibiotics such as tetracyclines, trimethoprim/sulfamethoxazole and penicillins is frequently observed in Ghana [8,9]. However, the underlying mechanisms of resistance in this region are not well understood and to our knowledge has not been described for *S. aureus* from chronic wounds.

Effective surveillance of antimicrobial resistance in bacteria including *S. aureus* is essential for estimating the burden of resistance and molecular strain typing provides important information for understanding the spread of resistant clones. However, in Africa both surveillance and strain typing information are scarce due to the limited diagnostic microbiology infrastructure generally available in large parts of the continent. Molecular typing including *spa*-typing, multi-locus sequencing and also whole genome sequence typing has been applied in only a few studies in *S. aureus* in humans and livestock in Ghana [3,10,11,12,13,14], identifying strains of multiple clonal clusters [14]. In particular, the MRSA clone sequence type (ST)88-IV (2B) is not only abundant in Ghana, but also in other African counties: Angola, Cameroon, Gabon, Madagascar, Nigeria, as well as São Tomé e Príncipe [15]. In Ghana, reported rates range between 24.2–83.3% of all MRSA isolates [12].

In a previous study *S. aureus* was isolated from 14.0% (*n* = 28) of samples from patients with chronic wounds in Ghana [5]. In that study, a high frequency of methicillin-resistance (29%) was noted. Moreover, resistance to other commonly used antibiotics like penicillins, tetracyclines and trimethoprim/sulfamethoxazole was frequently observed. For Ghanaian *S. aureus* isolates from wounds, data on prevalent clones, resistance mechanisms and pathogenicity-associated genetic determinants is limited. To fill this gap in the global epidemiological picture, we have subjected the 28 *S. aureus* isolates from our previous study to whole genome sequencing (WGS), aiming at analyzing the underlying molecular basis of antimicrobial resistance and the population structure of this strain collection.

## 2. Materials and Methods

### 2.1. Sample Collection, Microbiology and Antibiotic Susceptibility Testing

*S. aureus* was isolated from female and male patients ≥15 years with an infected wound at the Outpatient Department of the Agogo Presbyterian Hospital, in the Asante Akim North District of Ghana from January to November 2016. Sample collection and microbiological investigations were reported previously [5]. Antibiotic susceptibility was tested by the disk diffusion method and interpreted following the European Committee on Antimicrobial Susceptibility Testing (EUCAST) guidelines v.10.0 (http://www.eucast.org).

### 2.2. Whole Genome Sequencing and Data Analysis

Whole genome sequencing of the isolates was performed using the Illumina NextSeq platform. WGS data were analyzed using the Nullarbor pipeline (vers. 2.0.20181010; Seemann T, available at: https://github.com/tseemann/nullarbor). Reads were assembled with spades [16] (vers. 3.13.1) and annotated with Prokka [17] (vers. 1.13.3). The resistance and virulence gene profiles were determined with ABRicate (https://github.com/tseemann/abricate) (vers 0.9.9) employing NCBI AMR (7th October 2020; 5283 sequences), Resfinder [18] (3077 sequences) and VFDB [19] (2597 sequences) databases. SCCmec types were determined with the SCCmecFinder web tool (https://cge.cbs.dtu.dk/services/SCCmedFinder/10.1128/mSphere.00612-17).

The MLST sequence types were extracted from the WGS data using the MLST tool (vers. 2.16.1). Sequencing reads have been deposited in NCBI’s small reads archive (BioProject: PRJNA670821).

### 2.3. Ethical Considerations

The Committee on Human Research, Publications and Ethics, School of Medical Science, Kwame Nkrumah University of Science and Technology in Kumasi, Ghana, approved this study (approval number CHRPE/AP/078/16) on 14th December, 2015.

## 3. Results

The identified MLST sequence types and selected associated resistance and virulence genes are summarized in Table 1; the complete MLST, virulence factor and resistance gene datasets (VFDB, Resfinder and NCBI AMR), as well as AST results are available in the Appendix A (Appendix A) The most abundant sequence type was ST88 (8/28). All ST88 isolates were *mecA* positive, SCC*mec* type IV(2B), *agr* type 3 and negative for the Pantone–Valentine Leukocidin toxin (PVL) genes *lukS-PV/lukF-PV.* Seven of eight ST88 isolates carried the *fexB* gene, which confers resistance to chloramphenicol [20]. The five isolates of the second most abundant clone ST152 were all *mecA* negative and carried the PVL genes *lukS-PV/lukF-PV.*

Highest rates of phenotypic antimicrobial resistance were detected for penicillin (100%, 28/28), tetracycline (57%, 16/28) and trimethoprim/sulfamethoxazole (39%, 11/28) (Table 2). All isolates were susceptible to linezolid, rifampicin, fosfomycin, tigecycline, ciprofloxacin, levofloxacin and daptomycin. WGS resistance gene profiling identified corresponding acquired resistance genes in 97% (65/67) of all phenotypically detected antimicrobial resistances (Table 2). In all but one penicillin-resistant isolate penicillinase-encoding *blaZ* was detected. The single penicillin resistant, *blaZ* negative strain carried *mecA*, reasonably explaining the observed betalactam-resistant phenotype. All oxacillin-resistant isolates were *mecA* positive and belonged to ST88. M*ecC* was not detected in the strain collection. Two main mechanisms of resistance to tetracyclines have been described in *S. aureus*: active efflux, resulting from plasmid-located *tetK* and *tetL* genes and ribosomal protection mediated by *tetM* or *tetO* genes. In our collection tetracycline resistant isolates carried either tetK (7/16) alone or *tetL* (8/16) in combination with *tetM* (8/16). The *tetL* and *tetM* genes were exclusively detected in the ST88 MRSA isolates, while tet*K* was found in various clonal backgrounds. Beside mutation of the chromosomal dihydrofolate reductase (DHFR) gene, three acquired dihydrofolate reductase gene variants are known to confer resistance to trimethoprim in *S. aureus* of human origin: *dfrA*, *dfrG* and *dfrK*. In our collection all trimethoprim/sulfamethoxazole resistant isolates carried *dfrG*. Of the two erythromycin resistant isolates one carried *msrA* and one *ermC*. As expected, the latter isolate was also resistant to clindamycin. Gentamicin resistance is most commonly conferred by aminoglycoside-modifying enzymes. However, in the single case of a gentamicin-resistant isolate no corresponding resistance gene was found.

## 4. Discussion

In this study we describe the molecular epidemiology of *S. aureus* isolated from chronic infected wounds in outpatients in a rural area of Ghana.

Overall, the frequency of MRSA (29%) was high, comparable to other clinical studies conducted in Ghana [6,21,22]. Nevertheless, the frequencies of MRSA seen in patients in Ghana seem subject to geographical variations [5,6,7]. Moreover, the previously reported high rates of resistance to orally available antibiotics including penicillin, tetracycline and cotrimoxazole in *S. aureus* [9] were confirmed by our study results. The particularly high frequencies of penicillin resistance might be attributed to the fact that penicillin-based antibiotics are amongst the most frequently prescribed drugs in Ghana and available over the counter without prescription [23]. High rates of resistance inevitably reduce effective antibiotic treatment in areas where resources are scarce. This favors the use of cleaning and disinfecting procedures for the management of wound infections whenever clinically possible.

All isolated MRSA belonged to ST88, described as the dominant clone in various African countries, including Ghana [15]. In addition, all but one MRSA strain also carried the *fexB* gene, conferring resistance to chloramphenicol, which was not found in any of the other sequence types. Previously, *fexB* has only been described in *S. aureus* strains from Ghanaian patients with Buruli ulcer [20], also a chronic wound. As previously described in 2007 [24], chloramphenicol has been extensively prescribed in Africa, although a low risk of 0.002% for chloramphenicol-induced aplastic anemia had been described in Nigeria as early as in 1993 with an associated recommendation for strict risk-benefit-assessments prior to its prescription [25]. To the authors best knowledge, little has changed in the meantime and the substance is still in broad use in Sub-Saharan Africa, as it is readily available and shows excellent penetration even in difficult to reach compartments including bradytroph tissue like bone [26]. It is therefore possible that that frequent application of chloramphenicol in patients that did not respond to beta-lactam antibiotics, due to *mecA*-carriage of their *S. aureus* strains, might have facilitated the selection of *fexB* carrying bacteria. Due to the lack of reliable clinical data on previous antibiotic treatment this hypothesis could not be confirmed. Other mechanisms of resistance detected included *dfrG*, conferring resistance to trimethoprim, frequently found in strains isolated from Ghana [27,28]. Earlier this was regarded as an infrequent cause of trimethoprim resistance in *S. aureus* isolated from patients but is now widespread in Africa and common in *S. aureus* from ill travelers returning to Europe [27].

Panton-Valentine Leukocidin (PVL), which has been proposed as an epidemiological marker for severe skin infections [29], was encoded in strains of the ST121 and ST152 clonal lineages. The ST152 clonal lineage, in particular, is both known to be associated with PVL expression [30] and wide distribution in Ghana [10,31].

## 5. Conclusions

This study provides insight into the molecular epidemiology of *S. aureus* sequence types found in chronic infected wounds in a rural area of Ghana.

However, the number of samples used was quite small, and they were taken from outpatients in one hospital, so they may not be representative of the community or the wider area. Moreover, we did not have reliable information about prior use of antibiotics in these patients.

Nevertheless, this study stipulates valuable information for understanding the spread of resistant clones found in patients visiting the study hospital, which is important for effective surveillance of antibiotic resistant *S. aureus* and vital for estimating the burden of resistance.

## Figures and Tables

**Table 1 microorganisms-08-02052-t001:** Sequence types and associated virulence and resistance genes.

Sequence Type	n	*lukS-PV/lukF-PV*	*mecA*	*fexB*
ST88	8	0	8	7
ST152	5	5	0	0
ST15	3	0	0	0
ST1	2	0	0	0
ST5	2	0	0	0
ST45	2	0	0	0
ST2434	2	0	0	0
ST72	1	0	0	0
ST121	1	1	0	0
ST3248	1	0	0	0
ST3249	1	0	0	0
Totals	28	6	8	7

**Table 2 microorganisms-08-02052-t002:** Phenotypic antibiotic resistance and associated genetic resistance markers.

Antibiotic	Phenotypic AST Resistant n (%)	WGS Resistance Gene or Mutation	Positive n (%)
Penicillin	28 (100)	*blaZ*	27 (96)
		*blaZ* neg*, mecA* pos	1 (4)
Tetracycline	16 (57)	*tetK*	7 (44)
		*tetL*	8 (50)
		*tetM*	8 (50)
Trimethoprim/ Sulfamethoxazole	11 (39)	*dfrG*	11 (100)
Oxacillin	8 (29)	*mecA*	8 (100)
Erythromycin	2 (7)	*msrA*	1 (50)
		*ermC*	1 (50)
Clindamycin	1 (4)	*ermC*	1 (100)
Gentamicin	1 (4)	not detected	none

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
