# Peer review of "Molecular Characterization of Staphylococcus aureus Isolated from Chronic Infected Wounds in Rural Ghana"

_microorganisms, 2020, doi:10.3390/microorganisms8122052_

Round 1

Reviewer 1 Report

Wolters et al are presenting short communication that is following their previous study on the pathogens from chronically infected wounds in patients from rural district in Ghana.  Authors performed WGS analysis of the S. aureus isolates (n=28) from the previous study. They determined MLST, presence of resistance and virulence genes for the strains. All MRSA (8/28) belongs to prevalent African lineage ST88 IV, and most frequent MSSA lineage is ST152 (PVL+). Manuscript contains also supplementary data that presenting results for virulence and resistance gene presence.

Weakness is that the study contains limited number of isolates, it is single centre study, and did not found anything unexpected or really new. There are also no information about patients (gender, age,…). Most of the findings just confirmed results of previous studies from Ghana or other African countries. Strength of the manuscript is however that WGS gives deeper insight and could be of use to other researchers interested in S. aureus in Africa.

Specific comments:

Line 43. Please add a reference.

Line 44: reference no. 1 is review of studies that were published up to 2011, however in majority of them the isolates were collected much earlier. So the reference could not be used solely to describe increase of the MRSA prevalence in last 20 years. Please use more relevant references or change the sentence to reflect the content of the reference: e.g. “Review of the studies observing MRSA prevalence published up to 2011 indicated increasing prevalence of MRSA across the Africa at that time. There are no newer data.”

Line 45-46. Please be more specific with the results of the studies. What were the geographic differences?

Lines 52-56: It is rather weird that in one sentence authors put that surveillance and strain typing data are scarce in Ghana/Africa... and in following sentence that: there is number of studies in Ghana including those performing WGS. Both could not be true.

Materials and methods

Line 78-84 if there is no limit to number of references, please, appreciate the work of the authors of the WGS analysis tools by providing references to their work.

Results

In addition to MLST, for MRSA, that are all the same ST88 lineage, some other more discriminative method should be used to assess their relationship e.g. whole or core genome MLST.

Line 99 to 120: this paragraph repeats results from Table 2 to such extend that the table itself is no more needed. Authors should either remove the table or adjust the text.

Supplementary data:

Most of the data is actually presented in the supplementary data, but there is no legend or caption to explain e.g. what is the meaning of numbers that are presented in the table.

I guess 100.00 means 100% identity to the gene in question, but what does it mean when there are two values for the same gene? E.g. tet(K) in NCBI AMR results.

Results of AST and genes found by Resfinder and NCBI AMR should be combined into one table. It will be much clearer to see genes next to AST result for corresponding antimicrobials.

Discussion

Line 136: Authors speculates that chloramphenicol resistance in MRSA strains from chronic wounds in Africa is selected due to extensive use of chloramphenicol as the alternative to beta lactams in treatment of MRSA infection. Nowadays in general chloramphenicol is not important antimicrobial in treatment of staphylococcal or other infections. I am missing any reference to clarify this statement. At least whether the use of chloramphenicol is common practise in treating staphylococcal/MRSA infection in Ghana/Africa.

Author Response

thank you very much for your valuable comments and suggestions. Please find below (upload) our answers reflecting what was changed in the document.

We are hoping that we sufficiently provided what was suggested.

Please let us know if there is anything you disagree on or needs further changing.

best wishes and merry xmas,

Denise Dekker

Reviewer 2 Report

This study describes molecular epidemiological information of 28 S. aureus isolates from outpatients visited to a hospital in Ghana. This is one of the welcomed studies increasing our understanding of antimicrobial resistance situation, and valuable to be published. Since they submitted all the strains to whole genome sequencing, this reviewer would like to ask authors to add a bit more information, since readers would have interests.

Major

  1. SCC type, agr type, etc are also valuable information in staphylococcal epidemiology. Please add these information. Did all of the MRSAs have same SCC sequence?
  2. Please add the location of resistance genes (chromosome or plasmid or transposon etc) as much as possible, though it might depends on the quality of sequences.

Minor

Penicillin resistance was impressive. beta-lactams are OTC in this country? How often do people take it previously and currently? Please add some more discussion about the possible situation that led to this resistance rate.

Author Response

thank you very much for your valuable comments and suggestions. We have changed the document accordingly and are hoping that we fulfilled what was required.

Please let us know if you disagree or have further suggestions. Please find our answers in the uploaded document.

Best wishes and merry xmas,

Denise Dekker
